# Cardiorespiratory fitness in late adolescence and long-term risk of psoriasis and psoriatic arthritis among Swedish men

**Marta Laskowski**[1,2]*, **Linus Schiöler**[3], **Helena Gustafsson**[2,4], **Ann-Marie Wennberg**[1,2], **Maria Åberg**[5,6], **Kjell Torén**[3]

1 Department of Dermatology and Venereology, Institute of Clinical Sciences, The Sahlgrenska Academy, University of Gothenburg, Gothenburg, Sweden, 2 Department of Dermatology and Venereology, Region Västra Götaland, Sahlgrenska University Hospital, Gothenburg, Sweden, 3 Occupational and Environmental Medicine, School of Public Health and Community Medicine, Institute of Medicine, The Sahlgrenska Academy, University of Gothenburg, Gothenburg, Sweden, 4 Department of Physiology, Institute of Neuroscience and Physiology, The Sahlgrenska Academy, University of Gothenburg, Gothenburg, Sweden, 5 School of Public Health and Community Medicine/Primary Health Care, Institute of Medicine, The Sahlgrenska Academy, University of Gothenburg, Gothenburg, Sweden, 6 Region Västra Götaland, Regionhälsan, Gothenburg, Sweden

* marta.laskowski@vgregion.se

**Data Availability Statement:** Data cannot be shared publicly since the original approval by the Swedish Ethical Review Board does not include such a direct, free access. Since information about

## Abstract

### Background

Psoriasis is a chronic immune-mediated disease and psoriatic arthritis is a common coexisting condition. Cardiorespiratory fitness is the overall capacity to perform exertion exercise. Low levels of cardiorespiratory fitness are associated with negative health outcomes. Individuals with psoriasis have lower cardiorespiratory fitness compared with individuals without psoriasis. There are no previous studies exploring the association between cardiorespiratory fitness and new-onset psoriasis and psoriatic arthritis.

### Methods

With the objective to investigate whether low cardiorespiratory fitness in late adolescence increases the risk for onset of psoriasis and psoriatic arthritis, a cohort of Swedish men in compulsory military service between 1968 and 2005 was created using data from the Swedish Military Service Conscription Register. Cardiorespiratory fitness, estimated by maximum capacity cycle ergometer testing at conscription, was divided into three groups: high, medium, and low. Diagnoses were obtained using the Swedish National Patient Register and cohort members were followed from conscription until an event, new-onset psoriasis or psoriatic arthritis, occurred, or at the latest until 31 December 2016. Cox regression models adjusted for confounders at conscription were used to obtain hazard ratios with 95% confidence intervals for incident psoriasis and psoriatic arthritis.

### Results

During the follow-up period (median follow-up time 31 years, range 0–48 years), 20,679 cases of incident psoriasis and 6,133 cases of incident psoriatic arthritis were found among

health is regarded as sensitive information, an approval from a Swedish Ethical Review Board is required when sharing such data, according to Swedish law. Requests for data access can be directed to; Kjell Torén, School of Public Health and Community Medicine, University of Gothenburg; kjell.toren@amm.gu.se. Readers who are interested in accessing direct data underlying the findings in the current study are referred to the Swedish Ethical Review Authority (address: Etikprövningsmyndigheten, Box 2110, 750 02 Uppsala, Sweden; e-mail: registrator@etikprovning.se, phone: 010-475 0800).

**Funding:** ML and HG report grants from The Psoriasis Fund (Psoriasisfonden, www.psoriasisforbundet.se), grants from The Sahlgrenska University Hospital Funds (SU-fonderna, SU-786171, www.researchweb.org), grants from The Wilhelm and Martina Lundgren Science Fund (Wilhelm och Martina Lundgrens Vetenskapsfond, 2019-3016, https://wmlundgren.se), grants from The Royal and Hvitfeldts' Foundation (Kungliga och Hvitfeldtska stiftelsen, www.hvitfeldtskastiftelsen.se), during the conduct of the study; LS and KT report grants from The Swedish Council for Working Life, Health, and Welfare (FORTE, https://forte.se), grants from The Swedish State under the ALF agreement between the Government of Sweden and the County Councils (ALFGBG-74570, www.alfvastragotaland.se), during the conduct of the study; A-MW and MÅ have nothing to disclose. The funders had no role in study design, data collection and analysis, decision to publish, or preparation of the manuscript.

**Competing interests:** The authors have declared that no competing interests exist.

1,228,562 men (mean age at baseline 18.3 years). There was a significant association between low cardiorespiratory fitness and incident psoriasis and psoriatic arthritis (hazard ratio 1.35 (95% confidence interval 1.26–1.44) and 1.44 (95% confidence interval 1.28–1.63), respectively).

## Conclusions

These novel findings suggest that low cardiorespiratory fitness at an early age is associated with increased risk of incident psoriasis and psoriatic arthritis among men, and highlight the importance of assessing cardiorespiratory fitness early in life.

## Introduction

Psoriasis is a chronic immune-mediated disease of the skin, with wide dermatological manifestations, most commonly involving erythematous, scaly plaques on the extensor surfaces of the joints [1]. A few decades ago, psoriasis was considered to exclusively affect the skin, but it has now been established that psoriasis is a multisystem disease associated with various comorbidities, including cardiovascular disease [2]. Psoriatic arthritis, a chronic inflammatory spondyloarthritis, is the most common coexisting condition affecting up to one-third of patients with psoriasis. In the majority of cases it is preceded by psoriasis for 10 years on average [3]. Psoriasis and psoriatic arthritis are considered to affect men and women, with no predilection for sex [4, 5].

Low levels of physical activity are a risk factor for mortality, morbidity and some autoimmune diseases [6]. The relationship between physical activity and psoriasis is sparsely studied. However, some studies suggest that high levels of physical activity decrease the risk for incident psoriasis in women [7, 8].

Cardiorespiratory fitness (CRF), which should not be confused with physical activity, or with exercise [9], is the overall capacity of the respiratory and cardiovascular system to perform continuous aerobic activity over a longer time period. Levels of CRF are to a limited extent related to the amount and intensity of physical activity, but are in large part determined by other factors, including heredity, which is considered to account for 45–50% of the response to physical activity [10]. High CRF levels are associated with positive effects on several aspects of health, including cardiovascular health, and all-cause mortality [11], independent of physical activity levels [12, 13]. Davidson et al. recently showed that physical activity, on the other hand, was no longer a predictor of mortality after controlling for CRF [14]. Adding to previous discrepant results [15–19], this suggests that higher CRF levels, rather than physical activity levels, are important for gaining health benefits. Cardiorespiratory fitness in relation to psoriasis is sparsely studied, but individuals with diagnosed psoriasis in a general American population had lower CRF compared with individuals without psoriasis, independent of physical activity levels, psoriasis severity, and body mass index (BMI) [20]. Considering the modest difference between the groups, the clinical significance of these results should, however, be considered as uncertain.

To our knowledge, there are no previous studies exploring the association between new-onset psoriasis and psoriatic arthritis in relation to CRF. In the present study, we set out to investigate whether objectively measured low CRF in late adolescence is associated with incident psoriasis and psoriatic arthritis among men. We therefore performed a prospective cohort study of all Swedish men who underwent physical examinations when enlisting for

compulsory military service at the age of 18 years. These men were followed for up to 48 years through linkage to the Swedish National Patient Register (NPR) which identified all cases of psoriasis and psoriatic arthritis.

## Patients and methods

### Study population

The Swedish Military Service Conscription Register covers the physical and psychological evaluations performed by physicians and psychologists on all Swedish conscripts prior to enlistment. By using this register, we identified a cohort of all young Swedish individuals enlisting for Swedish military service between 1968 and 2005 (n = 1,886,542). Men who were incarcerated or who had severe chronic somatic or mental conditions or disabilities were excluded from compulsory enlistment (c. 2–3% each year). Physical and psychological evaluations, details of which have been published previously [21, 22], were performed as part of the enlistment protocol. Conscripts with a reused personal identity number (n = 1,155), women (n = 10,228), conscripts aged <16 or >25 years (n = 54,718), conscripts with psoriasis or psoriatic arthritis diagnosis at conscription (n = 8,272), and conscripts with missing data on CRF (n = 582,937) or on conscription site (n = 670) were excluded, resulting in a cohort of 1,228,562 men included in this study (Fig 1).

### Cardiorespiratory fitness

Cardiorespiratory fitness was assessed by a bicycle ergometer test, with the final work rate (Wmax) divided by body mass, details of which have been described by Nordesjo and Schele [21]. Data were recorded as Wmax (1972–2005), Wmax/kg (1997–2005) or as fitness category (1968–2005). There were nine categories which we used as a measure of fitness, further categorizing them into "low" (score 1–3), "medium" (score 4–6) and "high" (score 7–9) [21, 23, 24]. The high fitness category was considered as the reference category for hazard ratio (HR) estimations. For the conscription years 2000–2005, the lowest three levels (1–3) were not recorded; therefore, during this period these levels were missing ("Missing data", Fig 1). Estimated values were excluded from the entire follow-up period as they were considered as missing data.

### Register data and classification of psoriasis and psoriatic arthritis

Psoriasis and psoriatic arthritis were identified in the NPR using the eighth to tenth revision of the International Classification of Diseases (ICD-8–ICD-10). All health care episodes in inpatient and outpatient specialist care are mandatorily reported to the NPR. During the study

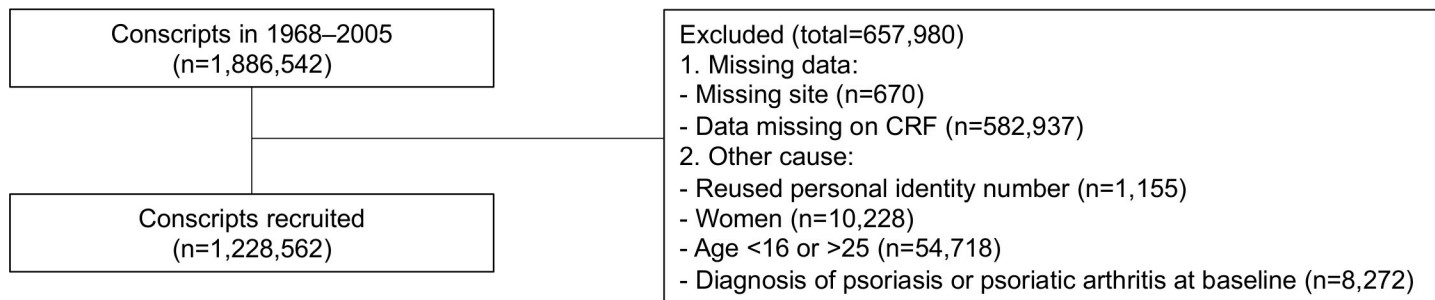

**Fig 1. Flowchart of included and excluded conscripts.** Number of included and excluded conscripts and reason for exclusion.

period, the coverage of the NPR, in terms of reporting hospitals and county councils, increased gradually, being nearly complete after 1987 and including diagnoses from hospital outpatient care from 2001 [25]. Using the Longitudinal Integration Database for Health Insurance and Labour Market Studies (Swedish acronym LISA), which includes all registered inhabitants from the age of 16 years, we obtained information on emigration and parental education. Parental education is divided into seven levels: <9 years, pre-high school education of 9 years, high school education (2 years), high school education (3 years), college/university (<3 years), college/university (≥3 years), and research education [26]. The Swedish Cause of Death Register [27] was used to obtain the date of death.

The principal and contributory diagnoses of psoriasis and psoriatic arthritis for all hospital and outpatient care in Sweden were obtained from the NPR. The ICD-8 was used from 1968 to 1986, the ICD-9 was used from 1987 to 1996, and thereafter the ICD-10 was used. Psoriasis of the skin was defined by ICD codes 696.10 and 696.19 (ICD-8), 696B (ICD-9), and L40.0–L40.4 and L40.8–L40.9 (ICD-10). The first diagnosis was accepted regardless of whether psoriasis was classified as a primary or secondary diagnosis. Psoriatic arthritis was defined as 696.00 (ICD-8), 696A (ICD-9) and L40.5 (ICD-10) and was included regardless of diagnostic position.

Data in the Swedish Military Service Conscription Register and the NPR were linked using the twelve-digit personal identification number allocated to all Swedish citizens. The linkage allowed the analysis of all reported cases of psoriasis and psoriatic arthritis during the follow-up period from 1968 to 31 December 2016.

Informed consent was not required as only de-identified and fully anonymized data were used. The study was approved by the Regional Ethical Review Board in Gothenburg (DNR 462–14, T252-18), and the approval included confirmation that informed consent was not required.

## Statistical analysis

The participants were followed from the date of conscription, hereafter referred to as the "index date", until either a first diagnosis of psoriasis or psoriatic arthritis, death from other causes, emigration from Sweden, or the end of follow-up, 31 December 2016.

Poisson regression was used to calculate incidence rates and corresponding 95% confidence intervals (CIs). Cox proportional hazards models were used to assess the effect of CRF on psoriasis and psoriatic arthritis. Age at conscription, year of conscription, and conscription test centre were included in all models. Additional models included alcohol abuse and diabetes mellitus at conscription (data obtained from the Swedish Military Service Conscription Register and the NPR), BMI at conscription (data obtained from the Swedish Military Service Conscription Register) and parental education (data obtained from LISA), all of which were considered as potential confounders. Alcohol abuse was defined by ICD codes 291 and 303 (ICD-8), 291, 303 and 305A (ICD-9) and F10 (ICD-10). Diabetes mellitus was defined by ICD codes 250 (ICD-8 and ICD-9) and E10–E14 (ICD-10). Restricted cubic splines with knots at the 5th, 35th, 65th, and 95th percentiles were used for BMI and conscription year. Age at conscription was adjusted as linear. Since the outpatient register was unavailable until 2001, we based the time scale on calendar year, and hence the major change in incidence is absorbed into the baseline hazard. We tested proportional hazards assumption using plots of Schoenfeld residuals and found a violation of the assumption for parental education, and test centre. Hence, we used stratified Cox models. Incidence rates among individuals with missing data on CRF were assessed in a post-hoc sensitivity analysis. A complementary analysis on the HRs for psoriasis and/or psoriatic arthritis was performed in conscripts with conscription dates

between 1995 and 2005 in order to address risk of bias due to short follow-up time. All statistical analyses were performed using SAS v 9.4 software (SAS Institute, Cary, NC, USA).

## Results

Table 1 shows the baseline characteristics of the study population stratified by CRF. Altogether 1,228,562 men (mean age 18.3±0.7 years) were included and participants were classified as having low (3.9%), medium (54%), or high (42%) CRF.

During the follow-up period of 0–48 years (median 31 years), first onset of psoriasis and/or psoriatic arthritis, according to any of the listed ICD codes, occurred in 23,296 individuals. There were 20,679 and 6,133 cases of new-onset psoriasis and psoriatic arthritis, respectively. Psoriasis and psoriatic arthritis co-occurred in 3,516 individuals. Diagnoses were listed as primary in more than 80% of cases. Persons diagnosed with new-onset psoriatic arthritis carried a previous diagnosis of psoriasis in 57% of cases.

As diagnoses from hospital outpatient care were integrated in the NPR in 2001, there was a considerable increase in reported cases from 2001; hence, incidence rates for 1968–2000 are presented in Table 2, and for 2001–2016, in Table 3. The incidence of psoriasis and/or psoriatic arthritis, according to the listed ICD codes, among the group with the lowest CRF was 167.1/ 100,000 person-years (PYs) (Table 3). The incidence of first diagnosis of psoriasis in the lowest CRF group was 149.0/100,000 PYs, while the incidence of psoriatic arthritis was 45.3/100,000 PYs (Table 3).

As shown in Table 4, there was evidence of a dose-response relationship, with increased risk of psoriasis and psoriatic arthritis being associated with lower CRF. The HR for psoriasis and/or psoriatic arthritis in any diagnostic position between 1968 and 31 December 2016, for the lowest CRF category, was 1.34 (95% CI 1.26–1.43) after full adjustment in model 4 (Table 4) compared with the highest CRF group. The HR of first diagnosis of psoriasis was 1.35 (95% CI 1.26–1.44), while the HR of psoriatic arthritis was 1.44 (95% CI 1.28–1.63) after full adjustment in the lowest CRF groups compared with the highest CRF groups (Table 4).

**Table 1. Baseline data of the study population.**

| Parameters | All | Cardiorespiratory fitness | | | Median age (years) at diagnosis (IQR) |
|---|---|---|---|---|---|
| | (N = 1,228,562) | High (7–9) *Reference* (n = 518,792) | Medium (4–6) (n = 661,622) | Low (1–3) (n = 48,148) | |
| Age (years) | 18.3±0.66 | 18.3±0.57 | 18.3±0.70 | 18.5±0.97 | |
| Height (m) | 1.80±0.07 | 1.80±0.06 | 1.80±0.07 | 1.80±0.07 | |
| Weight (kg) | 70.0±10.2 | 72.1±8.36 | 68.5±10.4 | 67.2±17.5 | |
| Body mass index (kg/m$^2$) | 21.7±2.77 | 22.1±2.26 | 21.5±2.89 | 21.5±4.84 | |
| Parental education (highest achieved level) | | | | | |
| 1–2 | 362,844 (30.6%) | 139,541 (27.8%) | 203,001 (31.9%) | 20,302 (44.8%) | |
| 3–4 | 506,302 (42.8%) | 203,182 (40.5%) | 284,460 (44.6%) | 18,660 (41.2%) | |
| 5–7 | 315,003 (26.6%) | 159,038 (31.7%) | 149,631 (23.5%) | 6,334 (14.0%) | |
| Psoriasis and/or psoriatic arthritis | 23,296 (1.9%) | 9,034 (38.8%) | 13,036 (56.0%) | 1,226 (5.3%) | 45.0 (37.0–51.0) |
| Psoriasis | 20,679 (1.7%) | 7,968 (38.5%) | 11,621 (56.2%) | 1,090 (5.3%) | 45.0 (37.0–52.0) |
| Psoriatic arthritis | 6,133 (0.5%) | 2,386 (38.9%) | 3,408 (55.6%) | 339 (5.5%) | 45.0 (39.0–51.0) |

Baseline data, psoriasis and psoriatic arthritis diagnosis by cardiorespiratory fitness (CRF) category and age at first psoriasis and psoriatic arthritis diagnosis in 1,228,562 male conscripts. For detailed information on parental education levels, please see "Register data and classification of psoriasis and psoriatic arthritis". Unless otherwise specified, data are presented as means ± standard deviation (SD) or n (%). IQR = interquartile range.

**Table 2. Incidence of psoriasis and psoriatic arthritis by cardiorespiratory fitness (CRF) category: High, medium or low.**

| Diagnosis | All | High (7–9) | Medium (4–6) | Low (1–3) |
|---|---|---|---|---|
| **Psoriasis and/or psoriatic arthritis** | | | | |
| N | 871 | 324 | 489 | 58 |
| Cases per 100,000 PYs | 4.27 | 3.68 | 4.60 | 6.03 |
| | (4.00–4.56) | (3.30–4.10) | (4.21–5.03) | (4.66–7.80) |
| **Psoriasis** | | | | |
| N | 662 | 236 | 339 | 47 |
| Cases per 100,000 PYs | 3.05 | 2.68 | 3.19 | 4.89 |
| | (2.82–3.30) | (2.36–3.05) | (2.87–3.55) | (3.67–6.50) |
| **Psoriatic arthritis** | | | | |
| N | 310 | 109 | 182 | 19 |
| Cases per 100,000 PYs | 1.52 | 1.24 | 1.71 | 1.97 |
| | (1.36–1.70) | (1.03–1.49) | (1.48–1.98) | (1.26–3.10) |

Incidence of psoriasis and psoriatic arthritis by category of cardiorespiratory fitness. Data is presented as crude incidence rates (95% CI). PYs = person-years.

*Table 2 represents crude incidence rates for 1968–2000, and Table 3 for 2001–2016.

Complementary analyses were performed on 582,937 subjects with data missing on CRF (S1 and S2 Tables), suggesting slightly higher incidence rates prior to 2000 and slightly lower incidence rates between 2001 and 2016 compared with the included individuals. Median follow-up time for psoriasis and/or psoriatic arthritis was 19 years in this group compared with 31 years among included individuals. The prevalence of psoriasis and/or psoriatic arthritis among the 1,228,562 included men was 1.9%, compared with 1.6% among those with missing data on CRF. The prevalence of psoriasis and psoriatic arthritis was 1.7% and 0.5%, respectively, among included individuals, compared with 1.5% and 0.4% among those excluded because of missing CRF values. Additional analysis resulted in HR 1.34 (95% CI 0.83–2.16) for psoriasis and/or psoriatic arthritis among participants with index dates after 1995 (S3 Table) compared with 1.34 (95% CI 1.26–1.43) among all participants (Table 4).

**Table 3. Incidence of psoriasis and psoriatic arthritis by cardiorespiratory fitness (CRF) category: High, medium or low.**

| Diagnosis | All | High (7–9) | Medium (4–6) | Low (1–3) |
|---|---|---|---|---|
| **Psoriasis and/or psoriatic arthritis** | | | | |
| N | 22,425 | 8,710 | 12,547 | 1,168 |
| Cases per 100,000 PYs | 125.3 | 116.0 | 129.4 | 167.1 |
| | (123.6–126.9) | (113.6–118.5) | (127.1–131.6) | (157.8–177.0) |
| **Psoriasis** | | | | |
| N | 20,057 | 7,732 | 11,282 | 1,043 |
| Cases per 100,000 PYs | 111.9 | 102.9 | 116.2 | 149.0 |
| | (110.3–113.4) | (100.6–105.2) | (114.0–118.3) | (140.2–158.3) |
| **Psoriatic arthritis** | | | | |
| N | 5,823 | 2,277 | 3,226 | 320 |
| Cases per 100,000 PYs | 32.3 | 30.1 | 33.0 | 45.3 |
| | (31.5–33.1) | (28.9–31.4) | (31.9–34.2) | (40.6–50.6) |

Incidence of psoriasis and psoriatic arthritis by category of cardiorespiratory fitness. Data is presented as crude incidence rates (95% CI). PYs = person-years.

*Table 2 represents crude incidence rates for 1968–2000, and Table 3 for 2001–2016.

**Table 4. Hazard ratios (HRs) for psoriasis and psoriatic arthritis in relation to cardiorespiratory fitness (CRF).**

| Diagnosis | Number (events/population) | Hazard ratio (95% CI) per CRF category | | |
|---|---|---|---|---|
| | | High (7–9) *Reference* | Medium (4–6) | Low (1–3) |
| **Psoriasis and/or psoriatic arthritis** | | | | |
| Model 1 | 23,296/1,227,882 | 1.00 | 1.14 (1.11–1.18) | 1.32 (1.25–1.41) |
| Model 2 | 23,296/1,227,882 | 1.00 | 1.14 (1.11–1.18) | 1.32 (1.24–1.40) |
| Model 3 | 23,225/1,224,428 | 1.00 | 1.18 (1.15–1.21) | 1.36 (1.28–1.45) |
| Model 4 | 22,492/1,180,457 | 1.00 | 1.17 (1.14–1.21) | 1.34 (1.26–1.43) |
| **Psoriasis** | | | | |
| Model 1 | 20,679/1,227,882 | 1.00 | 1.15 (1.12–1.19) | 1.33 (1.25–1.42) |
| Model 2 | 20,679/1,227,882 | 1.00 | 1.15 (1.12–1.19) | 1.33 (1.25–1.42) |
| Model 3 | 20,611/1,224,428 | 1.00 | 1.19 (1.15–1.22) | 1.36 (1.27–1.46) |
| Model 4 | 19,942/1,180,457 | 1.00 | 1.18 (1.14–1.22) | 1.35 (1.26–1.44) |
| **Psoriatic arthritis** | | | | |
| Model 1 | 6,133/1,227,882 | 1.00 | 1.15 (1.09–1.21) | 1.37 (1.22–1.54) |
| Model 2 | 6,133/1,227,882 | 1.00 | 1.15 (1.09–1.21) | 1.37 (1.22–1.53) |
| Model 3 | 6,116/1,224,428 | 1.00 | 1.21 (1.14–1.28) | 1.48 (1.32–1.67) |
| Model 4 | 5,945/1,180,457 | 1.00 | 1.19 (1.13–1.26) | 1.44 (1.28–1.63) |

Model 1: Adjusting for age at conscription, year of conscription, conscription test centre.

Model 2: Additionally adjusting for alcohol abuse and diabetes mellitus at conscription.

Model 3: Additionally adjusting for body mass index (BMI) at conscription.

Model 4: Additionally adjusting for parental education.

Hazard ratios (95% confidence intervals (CIs)) for psoriasis and psoriatic arthritis in relation to CRF.

## Discussion

The main findings of this prospective cohort study are that low levels of CRF in adolescence are a risk factor for psoriasis and psoriatic arthritis in men. This study presents unique data on the impact of objectively measured CRF on new-onset psoriasis and psoriatic arthritis, and adds psoriasis and psoriatic arthritis to the list of diseases associated with low CRF levels.

Physical inactivity is considered by some to be the fourth leading cause of death worldwide [28]. There is a well-known, overall positive effect of physical activity on autoimmunity, which is thought to be caused by an intricate sequence of events affecting large parts of the immune system, including the adaptive and innate immune system, cytokines, and hormone levels [6]. Research, predominantly performed on women, has shown protective effects of physical activity on new onset of several autoimmune conditions [6]. Lower levels of physical activity are reported among patients suffering from autoimmune diseases, including psoriasis [6, 29]. Although there are suggested associations between physical activity levels and psoriasis prevalence, incidence and severity, previous studies are limited by methodological issues [29]. One of these, a large prospective, questionnaire-based study of female US nurses, showed a decreased risk for incident psoriasis among nurses with high self-reported vigorous physical activity (adjusted relative risk (RR) 0.73 (95% CI 0.60–0.90)) during 14 years of follow-up [7]. Weaknesses in the study were that both psoriasis diagnoses and physical activity levels were self-reported [7]. A prospective questionnaire- and register-based cohort study in elderly white US women, performed by Prizment et al., supports these findings, showing a lower risk of incident psoriasis among women involved in regular physical activity (adjusted RR 0.80 (95% CI 0.70–1.0)) during 1991–2004 [8].

Increased physical activity levels are associated with better CRF, but only to a limited extent [15, 30]. Cardiorespiratory fitness is largely affected by hereditary factors [11, 31] and the

beneficial effects on mortality and morbidity are independent of levels of physical activity [12, 13]. Low CRF has been established as a risk factor for all-cause mortality [11], the mechanisms of which are thought to be multifactorial including, but not limited to, levels of aerobic exercise [32, 33]. Furthermore, it was recently shown that physical inactivity did not remain a risk factor for mortality when correcting for CRF status [14]. Hence CRF, which offers an objective measurement of cardiovascular function, seems to be a more important risk predictor for health events compared with physical activity.

The mechanisms behind the beneficial effects of high CRF are not fully understood. Altered fat distribution, reduced inflammatory burden and blood pressure, as well as advantageous changes in blood lipids and blood glucose, have been suggested as possible mechanisms of the protective effects of high CRF levels [32]. Although an association between high CRF and lower BMI has been suggested in a general population [34], CRF levels were recently shown to be lower among psoriasis patients, which could not be explained by BMI [20]. As decreased parasympathetic and increased sympathetic activity is associated with reduced CRF in other populations [35], autonomic nervous system dysfunction has been suggested as a potential mechanism driving low CRF in psoriasis patients [20, 36, 37].

Considering the well-known risk for cardiovascular disease in psoriasis patients [2], the association between low CRF and enhanced risk of cardiovascular disease, and the reduced mortality risk due to improved CRF [11, 12], there are important health benefits to be gained from identifying low CRF levels. The results from the current study indicate that low levels of CRF in late adolescence could also be a possible predictor of psoriasis and psoriatic arthritis, and underline the importance of assessing CRF levels early in life.

## Strengths and limitations

The sample size and large numbers of incident cases are strengths of the current study. The risk of measurement errors was reduced by direct measurement of maximum work capacity, i.e. CRF. The high coverage and validity of the registers used are further strengths of the study. The Swedish National Inpatient Register (IPR) is part of the NPR, which currently covers almost 100% of all hospital care and about 80% of all outpatient care. It was launched in 1964 and had achieved almost complete coverage by 1987, in terms of reporting hospitals and county councils. The main misclassification errors in the IPR occur because of diagnostic errors, translation errors and coding errors, and diagnoses are generally assumed to have higher validity in patients with severe disease than in those with mild disease. The exclusion of primary care diagnoses could, hence, reduce the risk of invalid classification due to diagnostic difficulties in the assessment of mild disease. As diagnoses from hospital outpatient care have been included since 2001, data prior to 2001 underreport mild disease. When comparing the incidence data (Tables 2 and 3), low CRF was associated with an increased risk of psoriasis and psoriatic arthritis both prior to and after 2001, indicating that low CRF is associated with an increased risk independently of the severity of the disease.

Diagnoses in the NPR were not formally validated, since their main purpose is to be used for administrative purposes. However, the diagnoses of psoriasis and psoriatic arthritis, classified according to the ICD-10, have previously been shown to have high validity, with correctly recorded codes in at least 81% (psoriasis) and 63% (psoriasis and psoriatic arthritis) of cases, in Swedish data between 2005 and 2010 [38]. Diagnostic codes of psoriasis and psoriatic arthritis classified in the ICD-9 had high sensitivity in American data between 1996 and 2009 [39], but ICD-9 and Hospital Adaptation of the International Classification of Diseases (HICDA) (a modification of the ICD-8) psoriasis codes studied between 1976 and 2000 showed lower validity [40], indicating a risk of misclassification of the diagnoses from the earlier years also in

the current study. Misclassification bias should also be considered because of unavailability of data on psoriasis and psoriatic arthritis diagnosis at inclusion in the study.

We were able to adjust for obesity and other well-known risk factors for incident psoriasis and psoriatic arthritis [41, 42], but we had no access to data on other separate factors comprising metabolic syndrome, which have previously been linked to an increased risk of psoriasis [41]. However, the risk of confounding should be low, considering that the increased risk for psoriasis associated with high triglycerides, high-density lipoprotein (HDL) cholesterol and total cholesterol is decreased when adjusting for BMI [41]. Smoking and alcohol use have been associated with psoriasis and psoriatic arthritis [2, 43, 44]. Alcohol use was adjusted for in the analysis. However, the lack of direct data on smoking habits, in particular current tobacco use, is a weakness of the study, which could be a risk for residual confounding. Notably, educational levels are known to be linked to smoking status in northern Europe, including Sweden [45], which was adjusted for by using data on parental education. The effect of socioeconomic status on incident psoriasis and psoriatic arthritis is somewhat uncertain, but single studies indicate that high socioeconomic status may constitute a risk factor for psoriatic arthritis [46]. Diabetes mellitus was adjusted for as a baseline comorbidity. Other comorbidities and medications, which were not adjusted for could, however, present a risk of confounding. Since we had no access to data on physical activity, it was not possible to adjust for increased physical activity levels, which is a limitation of the study. Furthermore, missing data on any of the confounders could also constitute risk of bias.

Since there are indications that there has been an overall decline in mean CRF over the last decades, especially among men [47], the long follow-up time may constitute a limitation due to the considerable time interval between the single measurement of CRF and the outcome measures. We addressed this issue by performing additional analysis on individuals with index dates after 1995 (S3 Table), which showed no difference in HR for psoriasis and/or psoriatic arthritis compared with all participants (Table 4).

An additional sensitivity analysis was performed on individuals with missing data on CRF (n = 582,937), indicating younger age at diagnosis, slightly higher BMI, and higher parental education levels among individuals with missing data. Incidence rates were slightly higher prior to 2000 and slightly lower between 2001 and 2016 in the group with missing data on CRF (S2 Table) compared with the included individuals. Furthermore, the median conscription year for the included individuals was 1981, compared with 1992 for those with missing data, which may in part explain the discrepancies.

## Conclusions

Our results offer unique prospective data on the association between objectively measured low CRF and higher incidence of psoriasis and psoriatic arthritis in men. We found that low CRF at an early age is associated with increased risk of incident psoriasis and psoriatic arthritis. These results add to previous knowledge on the beneficial effects of high CRF and highlight the importance of assessing CRF early in life.

## Supporting information

**S1 Table. Baseline data on conscripts with missing data on cardiorespiratory fitness (CRF).** Baseline data and age at first psoriasis and psoriatic arthritis diagnosis in 582,937 male conscripts with CRF data missing. For detailed information on parental education levels, please see "Register data and classification of psoriasis and psoriatic arthritis". Unless otherwise specified, data are presented as means ± standard deviation (SD) or n (%).

IQR = interquartile range.
(DOCX)

**S2 Table. Incidence of psoriasis and psoriatic arthritis in conscripts with missing data on cardiorespiratory fitness (CRF) (n = 582,937).** Data are presented as crude incidence rates (95% confidence interval (CI)). PYs = person-years. S2A Table represents incidence rates for 1968–2000, and S2B Table for 2001–2016.
(DOCX)

**S3 Table. Hazard ratios (HRs) for psoriasis and/or psoriatic arthritis in men conscripted after 1995.** Hazard ratios (95% confidence intervals (CIs)) for psoriasis and/or psoriatic arthritis with respect to cardiorespiratory fitness (CRF) among those conscripted after 1995.
(DOCX)

## Acknowledgments

The researchers wish to thank all the study participants.

## Author Contributions

**Conceptualization:** Marta Laskowski, Linus Schiöler, Helena Gustafsson, Ann-Marie Wennberg, Maria Åberg, Kjell Torén.

**Data curation:** Linus Schiöler.

**Formal analysis:** Marta Laskowski, Linus Schiöler, Helena Gustafsson, Ann-Marie Wennberg, Maria Åberg, Kjell Torén.

**Funding acquisition:** Marta Laskowski, Helena Gustafsson, Ann-Marie Wennberg, Kjell Torén.

**Investigation:** Marta Laskowski.

**Methodology:** Marta Laskowski, Linus Schiöler, Helena Gustafsson, Ann-Marie Wennberg, Maria Åberg, Kjell Torén.

**Resources:** Helena Gustafsson, Kjell Torén.

**Software:** Linus Schiöler.

**Supervision:** Helena Gustafsson, Ann-Marie Wennberg, Kjell Torén.

**Writing – original draft:** Marta Laskowski.

**Writing – review & editing:** Linus Schiöler, Helena Gustafsson, Ann-Marie Wennberg, Maria Åberg, Kjell Torén.

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
