## [Decision Letter · Decision Letter 0]

28 Sep 2020

PONE-D-20-20129

Cardiorespiratory fitness in late adolescence and long-term risk of psoriasis and psoriatic arthritis among Swedish men

PLOS ONE

Dear Dr. Laskowski,

Thank you for submitting your manuscript to PLOS ONE. After careful consideration, we feel that it has merit but does not fully meet PLOS ONE’s publication criteria as it currently stands. Therefore, we invite you to submit a revised version of the manuscript that addresses the points raised during the review process.

We look forward to receiving your revised manuscript.

Kind regards,

Sreeram V. Ramagopalan

Academic Editor

PLOS ONE

Journal Requirements:

2. In your ethics statement in the manuscript and in the online submission form, please provide additional information about the patient records used in your retrospective study. Specifically, please ensure that you have discussed whether all data were fully anonymized before you accessed them and/or whether the IRB or ethics committee waived the requirement for informed consent. If patients provided informed written consent to have data from their medical records used in research, please include this information. Additionally, in your ethics statement please state that the Regional Ethical Review Board in Gothenburg (DNR 462-14, T252-18) confirmed that informed consent was indeed not required.

Reviewers' comments:

Reviewer's Responses to Questions

**Comments to the Author**

1. Is the manuscript technically sound, and do the data support the conclusions?

Reviewer #1: Yes

2. Has the statistical analysis been performed appropriately and rigorously? 

Reviewer #1: Yes

3. Have the authors made all data underlying the findings in their manuscript fully available?

Reviewer #1: No

4. Is the manuscript presented in an intelligible fashion and written in standard English?

Reviewer #1: No

5. Review Comments to the Author

Reviewer #1: Thank you very much for giving me the opportunity to review the manuscript: PONE-D-20-20129. In this study the authors investigated the association between cardiorespiratory fitness (CRF) in late adolescence and long-term risk of psoriasis and psoriatic arthritis among Swedish men using Swedish Military Service Conscription Register and linking it with other routinely collected national health data in Sweden. The study found that low cardiorespiratory fitness was associated with incident psoriasis and psoriatic arthritis.

This longitudinal cohort study has the advantage of large sample size from national database and a long follow-up. The manuscript provides new data and hypothesis. However, certain aspects of the study need further clarification and explanation to convey the results of the study clearly. I think there is potential to improve this paper with attention to some details that are mentioned below.

First, the reasoning for hypothesis that lower CRF increases the risk of psoriasis (Pso) and psoriatic arthritis (PsoA) is not clearly explained. In the Introduction section CRF is mentioned as having no relation with physical activity and in Discussion section (line 264) there seem to be an association between CRF and physical activity. I suggest adding clear hypothesis and plausible mechanism behind it including how physical activity and CRF are linked to each other. For example, is physical activity a proxy for CRF or the other way around or is CRF an intermediary in plausible causal pathway between physical activity and psoriasis?

Second, the study has major limitation due to the presence of time-varying nature of the exposure and confounders such as body mass index (BMI). Maybe it needs mention in the limitation section. Another limitation is the misclassification bias due differential data availability during the earlier years of the study compared to the later years. Please see below for detailed suggestions.

Third, information is available on a very few confounders in the study, data on other confounders such as comorbidities, lab values, medications etc are missing.

Finally, the manuscript needs to be revised to make it clearer. Please see my detailed comments for each section below:

Abstract:

• Line 32: “a cohort of Swedish men (mean age 18.3 years)” This fit more into Result section than in Methods section, because mean age was computed ‘after’ making the cohort.

• Line 40-41: This modeling information is best suited for Method section.

• Line 41-42: “During the follow-up period (0–48 years),” Please provide median follow-up time.

Introduction:

• Line 68-70: “To the authors’ knowledge, there are no prospective studies on the association between physical activity levels and incident psoriasis among men and the association between new-onset PsoA and physical activity levels has not been studied.” This is confusing, may be the authors should remove this, as the study is not examining this gap.

• What is the link between CRF and physical activity?

Patients and Methods:

Study population:

• Line 105, “Figure 1. Flowchart of included and excluded conscripts.” Figure 1 not only shows the selection process of the study population, it also has outcomes included in it. I suggest keeping outcome reserved for Table 1 in Result section.

Cardiorespiratory fitness:

• Line 114: “For conscription years 2000–2005, the lowest three levels (1–3) were not recorded, thus during this period these levels were missing. Estimated values were excluded from the entire follow-up period as they were considered as missing data.” Does this mean these subjects were excluded from the trial as there is no complete information on the exposure? if yes is it included in the exclusion group “Data missing on CRF” in Figure 1? Is it correct to assume that there is no subject from the year 2000-2005 due to this missing data?

Statistical analysis:

• Line 160: “Age at conscription was adjusted as linear.” Is age really a confounder here looking at inclusion of only adolescent between 16 and 25 years of age?

• Line 162: Did the author also validated the Cox model for any outliers and non-linearity?

• Line 164: “…though the change in hazard ratio (HR) for CRF was negligible”. These are results, I suggest removing it from this section.

• Line 164, “ In order to address risk of bias due to missing data and short follow-up time, complementary analyses were performed on subjects with missing data on CRF, and HRs for incident psoriasis and/or PsoA were assessed among participants with index dates after 1995.” This is little confusing to me, if the data on CRF was missing then what was the predictive variable for this analysis? Also, I understood that all subjects with missing data on CRF were excluded already. After reading results I understand that there were two complimentary/sensitivity analysis done. One was done among excluded group of subjects with missing data and only incidence rates were measured. Second analysis was a stratified analysis to estimate HRs based on years of inclusion from 1968-1995 and 1995-2005. Please rephrase to make it clear.

About ‘short follow up time’, up to 48 years of follow-up time is not really a short follow-up time, but I don’t know the median years of follow up as it is not mentioned in the manuscript. Other than that, the authors can perform sensitivity analysis by externally adjusting for missing variables (not for the exposure variable though).

• It is not mentioned anywhere in the Method section which category of CRF was used as reference category for HR estimations, it would be easier to read the paper if it is explicitly stated in the Method section.

Results

• Line 171: “Table 1 shows the baseline characteristics of the study population stratified by CRF.” This is incorrect statement, Table 1 shows baseline characteristics according to the outcome (Pso and PsoA). I suggest making necessary changes to the table to show characteristics according to the exposure.

• Line 174: The follow up period was not 48 years; the maximum possible follow-up period was 48 years. I suggest adding median follow up time instead.

• Line 180: Maybe it should be called ‘baseline’ data instead of ‘basic’ data.

• Line 190: “…after full adjustment,” I could not find in the Method section that the authors intended to adjust the incidence rates. It would be beneficial to explain this is the Methods section for which variable(s) the incidence rates were adjusted.

• Table 2, it is not mentioned anywhere that the incidence rates were adjusted as mentioned in the text in line 190. Please clarify.

• Table 3, page 12: I suggest changing the orientation of the table and have No. of events, High CRF, Med CRF, and Low CRF as columns; and have all the models as rows. This will maintain the consistency similar to Table 2 and readers will get ‘events per exposure’ categories, in current format the total no. of events doesn’t add much value.

• Line 233, I think it should be “CI” instead of “Cis”.

• Line 235: I would be interested to know median follow-up time in both groups (the study population and the excluded group) to understand and explain the difference in incidence rates.

• Supplementary Tables don’t have titles, please add.

• Table S3: Why the HRs decreased after adjusting for BMI for this population whereas HRs increased for full cohort as shown in Table3?

Discussion:

• Line 246: “The main findings of this prospective cohort study are that low levels of CRF in adolescence are a risk factor for psoriasis and PsoA in men. Previous findings in women show an association between low self-reported physical activity levels and incident psoriasis.” In the Introduction section the authors mention that physical activity should not be confused with CRF; the mention of physical activity here is adding to the confusion. May be the authors want to add the rationale of providing information on physical activity here. For example, do the authors consider physical activity as proxy indicator of high CRF or what is the relation between physical activity and CRF?

• Line 183: “The overall positive predicted value (PPV) for all diagnoses in the IPR is estimated to be 85–95%[18].” This statement is incorrect, the PPV range is for most diagnosis not all. For autoimmune diseases it is relatively high though and PPV was not measured for Psoriasis.

• Line 292: “Another possible diagnostic bias which should be considered is the financial inducement of listing secondary diagnoses.” Why is this a bias if diagnosis codes have very high validity as claimed earlier in this section?

• Line 305: “We were able to adjust for obesity and other well-known risk factors for incident psoriasis and PsoA[34, 35].” The referenced study (ref. 35) found positive association of waist circumference, triglycerides, HDL cholesterol, but data on these were not available in this study. Maybe it needs a mention as well that there is unmeasured confounding.

• Line 319: “Additional analysis on individuals born late in the study period was therefore performed due to the variation in follow-up time, showing slightly higher HRs for psoriasis and/or PsoA among participants with an index date after 1995, compared to all participants (S3 Table).” This is contradictory to what is mentioned in line 241-243 as the HRs are 1.34 for both.

• Line 319: “Additional analysis on individuals born late in the study period was therefore performed due to the variation in….” I don’t think anybody was born during the study period and birth year was not the criteria to enter the study. I suggest rephrasing the sentence.

• Line 324: “The risk of bias due to missing data was addressed in an additional subanalysis, executed on individuals with data missing on CRF (n=582,937),…” This is not entirely correct because we have already excluded those subjects then it can only cause issues with external validity of the study. Bias due to missing data will be due to differential missingness of information on variables such as BMI or parental education etc.

• Line 326: “Incidence rates were slightly higher prior to 2000 and slightly lower between 2001 and 2016 in the group with missing data on CRF (S2 Table) compared to the included individuals.” What is the explanation for this observation? For full cohort we have higher incidence based on availability of hospital outpatient record from 2001 onwards.

• I suggest adding discussion on missing data on variables such as BMI, parental education etc. How was data on alcohol abuse collected?

• I suggest adding discussion on misclassification bias due to unavailability of outcome data (Pso and PsoA) from hospital outpatient records at the index date. I assume many subjects were included as having no Pso/PsoA incorrectly due this missing information.

Conclusions:

• Line 332: “Our results offer unique prospective data on the association between objectively measured low physical fitness and higher incidence” In this manuscript the relation between CRF and physical activity has not been explained appropriately, neither in introduction nor in discussion. I suggest to clearly explain how these are linked.

Other comments:

• Line 141: twelve-digit personal identification number allocated to all Swedish citizens.

• The manuscript could benefit from minor language revision e.g. on page 11, line 220: “As shown in Table 3, there was evidence of a dose-response relationship, with increased risk of psoriasis and PsoA being associated with lower CRF.”

6. PLOS authors have the option to publish the peer review history of their article (what does this mean?). If published, this will include your full peer review and any attached files.

Reviewer #1: No

---

## [Author Response · Author response to Decision Letter 0]

11 Nov 2020

Dear Mr Ramagopalan,

We would like to thank you for giving us the opportunity to revise and resubmit our manuscript. Please find our comments below. For clarity, all responses regarding changes in the text have been highlighted below in grey (attached document). 

PONE-D-20-20129

Cardiorespiratory fitness in late adolescence and long-term risk of psoriasis and psoriatic arthritis among Swedish men

PLOS ONE

Dear Dr. Laskowski,

Thank you for submitting your manuscript to PLOS ONE. After careful consideration, we feel that it has merit but does not fully meet PLOS ONE’s publication criteria as it currently stands. Therefore, we invite you to submit a revised version of the manuscript

that addresses the points raised during the review process.

We look forward to receiving your revised manuscript.

Kind regards,

Sreeram V. Ramagopalan

Academic Editor

PLOS ONE

Journal Requirements:

We have revised the manuscript in accordance with PLOS ONE’s requirements. 

2. In your ethics statement in the manuscript and in the online submission form, please provide additional information about the patient records used in your retrospective study. Specifically, please ensure that you have discussed whether all data were fully anonymized before you accessed them and/or whether the IRB or ethics committee waived the requirement for informed consent. If patients provided informed written consent to have data from their medical records used in research, please include this information. Additionally, in your ethics statement please state that the Regional Ethical Review Board in Gothenburg (DNR 462-14, T252-18) confirmed that informed consent was indeed not required.

We have clarified that all data were fully anonymized; also, we have added that the Regional Ethical Review Board in Gothenburg have confirmed that informed consent was not required (please see the last paragraph of the Register data and classification of psoriasis and psoriatic arthritis section). 

We have added the following statement in the revised Cover letter: “According to Swedish law, sensitive patient information in de-identified data sets cannot be shared unless this has been approved by the Ethical Review Board. The original approval by the Ethical Review Board did not include direct and free access and therefore, data cannot be made freely available. Readers who are interested in accessing direct data underlying the findings in the current study are referred to the Swedish Ethical Review Authority. Address: Etikprövningsmyndigheten, Box 2110, 750 02 Uppsala, Sweden; e-mail: registrator@etikprovning.se, phone: 010-475 08 00.” 

We have added a statement explaining the Swedish legal restrictions regarding sharing de-identified data sets containing detailed and sensitive patient information. We have also provided contact information for the Swedish Ethical Review Committee (please see Cover letter). 

 

Reviewers' comments:

Reviewer's Responses to Questions

Comments to the Author

1. Is the manuscript technically sound, and do the data support the conclusions?

Reviewer #1: Yes

2. Has the statistical analysis been performed appropriately and rigorously? 

Reviewer #1: Yes

3. Have the authors made all data underlying the findings in their manuscript fully available?

Reviewer #1: No

We have added a statement explaining this in the Cover letter. 

4. Is the manuscript presented in an intelligible fashion and written in standard English?

Reviewer #1: No

The manuscript has been edited and changes have been made according to the presented suggestions.

5. Review Comments to the Author

Reviewer #1: Thank you very much for giving me the opportunity to review the manuscript: PONE-D-20-20129. In this study the authors investigated the association between cardiorespiratory fitness (CRF) in late adolescence and long-term risk of psoriasis and psoriatic arthritis among Swedish men using Swedish Military Service Conscription Register and linking it with other routinely collected national health data in Sweden. The study found that low cardiorespiratory fitness was associated with incident psoriasis and psoriatic arthritis.

This longitudinal cohort study has the advantage of large sample size from national database and a long follow-up. The manuscript provides new data and hypothesis. However, certain aspects of the study need further clarification and explanation to convey the results of the study clearly. I think there is potential to improve this paper with attention to some details that are mentioned below.

First, the reasoning for hypothesis that lower CRF increases the risk of psoriasis (Pso) and psoriatic arthritis (PsoA) is not clearly explained. In the Introduction section CRF is mentioned as having no relation with physical activity and in Discussion section (line 264) there seem to be an association between CRF and physical activity. I suggest adding clear hypothesis and plausible mechanism behind it including how physical activity and CRF are linked to each other. For example, is physical activity a proxy for CRF or the other way around or is CRF an intermediary in plausible causal pathway between physical activity and psoriasis?

Second, the study has major limitation due to the presence of time-varying nature of the exposure and confounders such as body mass index (BMI). Maybe it needs mention in the limitation section. Another limitation is the misclassification bias due differential data availability during the earlier years of the study compared to the later years. Please see below for detailed suggestions.

Third, information is available on a very few confounders in the study, data on other confounders such as comorbidities, lab values, medications etc are missing.

Finally, the manuscript needs to be revised to make it clearer. Please see my detailed comments for each section below:

We highly appreciate your extensive and constructive review of our manuscript and wish to thank you for taking the time to further improve the manuscript. 

We agree that the relation between CRF and physical activity was not made sufficiently clear in the original manuscript. Therefore, we have made extensive changes to the manuscript to clarify the relation between CRF and physical activity and the effects of CRF on psoriasis and psoriatic arthritis. 

Thank you for your valuable point on the time-varying nature of the exposure and confounders, and misclassification bias due to differential data availability during earlier years of the study. We agree that these are important limitations and have added this point to the discussion.

We have also extended the Strengths and limitations section with a discussion on the lack of other confounders, such as lab values, medications and comorbidities. 

Furthermore, we have taken the opportunity to clarify the manuscript, with the help of your valuable comments. With your kind suggestions, we feel the manuscript has been improved significantly. We present our comments below, and hope that you will agree with this view. 

 

Abstract:

• Line 32: “a cohort of Swedish men (mean age 18.3 years)” This fit more into Result section than in Methods section, because mean age was computed ‘after’ making the cohort.

We have changed this according to your suggestion and moved the information on the cohort to the Results section of the Abstract. 

• Line 40-41: This modeling information is best suited for Method section.

The modelling information has been moved to the Methods section of the Abstract.

• Line 41-42: “During the follow-up period (0–48 years),” Please provide median follow-up time.

We have added information about the median follow-up time (31 years) both in the Abstract and in the Results section.

Introduction:

• Line 68-70: “To the authors’ knowledge, there are no prospective studies on the association between physical activity levels and incident psoriasis among men and the association between new-onset PsoA and physical activity levels has not been studied.” This is confusing, may be the authors should remove this, as the study is not examining this gap.

We fully agree. We have removed this.

• What is the link between CRF and physical activity? 

We have clarified the link between physical activity and CRF.

Patients and Methods:

Study population:

• Line 105, “Figure 1. Flowchart of included and excluded conscripts.” Figure 1 not only shows the selection process of the study population, it also has outcomes included in it. I suggest keeping outcome reserved for Table 1 in Result section.

We have removed the outcomes from the Figure.

Cardiorespiratory fitness:

• Line 114: “For conscription years 2000–2005, the lowest three levels (1–3) were not recorded, thus during this period these levels were missing. Estimated values were excluded from the entire follow-up period as they were considered as missing data.” Does this mean these subjects were excluded from the trial as there is no complete information on the exposure? if yes is it included in the exclusion group “Data missing on CRF” in Figure 1? Is it correct to assume that there is no subject from the year 2000-2005 due to this missing data?

Yes, this means that these subjects were excluded from the trial. Instead, they made up the group with “data missing on CRF”. No subjects from the years 2000-2005 were included. We have clarified this, with reference to Figure 1 (“Missing data”), in the Cardiorespiratory fitness section. 

Statistical analysis:

• Line 160: “Age at conscription was adjusted as linear.” Is age really a confounder here looking at inclusion of only adolescent between 16 and 25 years of age?

As your question suggests, age is most probably of minor importance. However, we have corrected for age as a confounder in our models in previous studies and would like to keep it as a confounding factor.

• Line 162: Did the author also validated the Cox model for any outliers and non-linearity?

Yes. Individuals with BMI between 15 and 60 are included in the model. Individuals with BMI below 15 and above 60 were considered as outliers and were excluded. Splines were used to account for non-linearity.

• Line 164: “…though the change in hazard ratio (HR) for CRF was negligible”. These are results, I suggest removing it from this section.

We have deleted this.

• Line 164, “ In order to address risk of bias due to missing data and short follow-up time, complementary analyses were performed on subjects with missing data on CRF, and HRs for incident psoriasis and/or PsoA were assessed among participants with index dates after 1995.” This is little confusing to me, if the data on CRF was missing then what was the predictive variable for this analysis? Also, I understood that all subjects with missing data on CRF were excluded already. After reading results I understand that there were two complimentary/sensitivity analysis done. One was done among excluded group of subjects with missing data and only incidence rates were measured. Second analysis was a stratified analysis to estimate HRs based on years of inclusion from 1968-1995 and 1995-2005. Please rephrase to make it clear.

Thank you, we have clarified this in the Statistics section.

About ‘short follow up time’, up to 48 years of follow-up time is not really a short follow-up time, but I don’t know the median years of follow up as it is not mentioned in the manuscript. Other than that, the authors can perform sensitivity analysis by externally adjusting for missing variables (not for the exposure variable though).

We have added information on the median follow-up time in the first paragraph of the Results section. “Short follow-up time” refers to the follow-up of individuals with conscription dates between 1995 and 2005 who were included in the complementary analysis; we have added their median follow-up. We have presented missing data on CRF in the last paragraph of the Results section.

• It is not mentioned anywhere in the Method section which category of CRF was used as reference category for HR estimations, it would be easier to read the paper if it is explicitly stated in the Method section.

We have now explicitly stated this in the Cardiorespiratory fitness (Methods) section and in the Tables. 

Results

• Line 171: “Table 1 shows the baseline characteristics of the study population stratified by CRF.” This is incorrect statement, Table 1 shows baseline characteristics according to the outcome (Pso and PsoA). I suggest making necessary changes to the table to show characteristics according to the exposure.

We have changed Table 1 and data now present characteristics of exposure. The same changes have been made in Supplementary Table 1 (Table S1). 

• Line 174: The follow up period was not 48 years; the maximum possible follow-up period was 48 years. I suggest adding median follow up time instead.

We have added this information to the sentence. 

• Line 180: Maybe it should be called ‘baseline’ data instead of ‘basic’ data.

We have changed this into “baseline data” in Tables 1 and S1 in accordance with your suggestion.

• Line 190: “…after full adjustment,” I could not find in the Method section that the authors intended to adjust the incidence rates. It would be beneficial to explain this is the Methods section for which variable(s) the incidence rates were adjusted.

• Table 2, it is not mentioned anywhere that the incidence rates were adjusted as mentioned in the text in line 190. Please clarify.

This is incorrect; thank you for making us aware of this mistake. Incidence rates were crude. We have corrected this.

• Table 3, page 12: I suggest changing the orientation of the table and have No. of events, High CRF, Med CRF, and Low CRF as columns; and have all the models as rows. This will maintain the consistency similar to Table 2 and readers will get ‘events per exposure’ categories, in current format the total no. of events doesn’t add much value.

We have changed Tables 3 and S3 according to the abovementioned suggestions.

• Line 233, I think it should be “CI” instead of “Cis”.

We have changed this.

• Line 235: I would be interested to know median follow-up time in both groups (the study population and the excluded group) to understand and explain the difference in incidence rates.

We have added the median follow-up time of both groups in the last paragraph of the Results section.

• Supplementary Tables don’t have titles, please add.

We have added titles for the Supplementary Tables. We have also changed Tables S1 and S3 according to your suggestions (characteristics of exposure (S1) and events per exposure category (S3)).

• Table S3: Why the HRs decreased after adjusting for BMI for this population whereas HRs increased for full cohort as shown in Table3?

This is an interesting observation. We have checked the data and this is correct. For some reason, the younger individuals (included in the group with a short follow-up time) had a different relation between BMI and CRF compared with those with a longer follow-up time. 

Discussion:

• Line 246: “The main findings of this prospective cohort study are that low levels of CRF in adolescence are a risk factor for psoriasis and PsoA in men. Previous findings in women show an association between low self-reported physical activity levels and incident psoriasis.” In the Introduction section the authors mention that physical activity should not be confused with CRF; the mention of physical activity here is adding to the confusion. May be the authors want to add the rationale of providing information on physical activity here. For example, do the authors consider physical activity as proxy indicator of high CRF or what is the relation between physical activity and CRF?

We have clarified the sections on physical activity and CRF.

• Line 183: “The overall positive predicted value (PPV) for all diagnoses in the IPR is estimated to be 85–95%[18].” This statement is incorrect, the PPV range is for most diagnosis not all. For autoimmune diseases it is relatively high though and PPV was not measured for Psoriasis.

Thank you for noticing this. We have removed this sentence and comment on the validity of psoriasis and PsoA diagnosis later on, in the Strengths and limitations section.

• Line 292: “Another possible diagnostic bias which should be considered is the financial inducement of listing secondary diagnoses.” Why is this a bias if diagnosis codes have very high validity as claimed earlier in this section?

After discussing this issue we have decided to remove this sentence. As you suggest, we do not consider this as a bias because of the high validity of the diagnosis codes.

• Line 305: “We were able to adjust for obesity and other well-known risk factors for incident psoriasis and PsoA[34, 35].” The referenced study (ref. 35) found positive association of waist circumference, triglycerides, HDL cholesterol, but data on these were not available in this study. Maybe it needs a mention as well that there is unmeasured confounding.

We have added this in the Strengths and limitations section. 

• Line 319: “Additional analysis on individuals born late in the study period was therefore performed due to the variation in follow-up time, showing slightly higher HRs for psoriasis and/or PsoA among participants with an index date after 1995, compared to all participants (S3 Table).” This is contradictory to what is mentioned in line 241-243 as the HRs are 1.34 for both.

Thank you, we have changed this.

• Line 319: “Additional analysis on individuals born late in the study period was therefore performed due to the variation in….” I don’t think anybody was born during the study period and birth year was not the criteria to enter the study. I suggest rephrasing the sentence.

We have clarified this.

• Line 324: “The risk of bias due to missing data was addressed in an additional subanalysis, executed on individuals with data missing on CRF (n=582,937),…” This is not entirely correct because we have already excluded those subjects then it can only cause issues with external validity of the study. Bias due to missing data will be due to differential missingness of information on variables such as BMI or parental education etc.

Thank you, we have rephrased this.

• Line 326: “Incidence rates were slightly higher prior to 2000 and slightly lower between 2001 and 2016 in the group with missing data on CRF (S2 Table) compared to the included individuals.” What is the explanation for this observation? For full cohort we have higher incidence based on availability of hospital outpatient record from 2001 onwards.

This is an interesting observation. Data from the outpatient register have been included since 2001 and hence, we see increased incidence rates in both groups after 2001, which is explained by the increased access to data. Rising incidence rates are seen also in the group with missing data on CRF, but for some reason, the increase induced by the availability of outpatient records from 2001 and onwards is smaller compared with the increase in the group of included individuals.

• I suggest adding discussion on missing data on variables such as BMI, parental education etc. How was data on alcohol abuse collected?

We have added detailed information on how data on confounders were collected, in the Statistical analysis section. We have also added a discussion on missing data on confounders, under Strengths and limitations. 

• I suggest adding discussion on misclassification bias due to unavailability of outcome data (Pso and PsoA) from hospital outpatient records at the index date. I assume many subjects were included as having no Pso/PsoA incorrectly due this missing information.

We have added this under Strengths and limitations.

Conclusions:

• Line 332: “Our results offer unique prospective data on the association between objectively measured low physical fitness and higher incidence” In this manuscript the relation between CRF and physical activity has not been explained appropriately, neither in introduction nor in discussion. I suggest to clearly explain how these are linked.

We have made changes in the Introduction, Discussion and Conclusions based on your suggestions to clarify the relation between CRF and physical activity.

Other comments:

• Line 141: twelve-digit personal identification number allocated to all Swedish citizens.

Language revision has been performed.

• The manuscript could benefit from minor language revision e.g. on page 11, line 220: “As shown in Table 3, there was evidence of a dose-response relationship, with increased risk of psoriasis and PsoA being associated with lower CRF.”

Language revision has been performed.

6. PLOS authors have the option to publish the peer review history of their article (what does this mean?). If published, this will include your full peer review and any attached files.

Do you want your identity to be public for this peer review? For information about this choice, including consent withdrawal, please see our Privacy Policy.

Reviewer #1: No

---

## [Editor Report · Decision Letter 1]

20 Nov 2020

Cardiorespiratory fitness in late adolescence and long-term risk of psoriasis and psoriatic arthritis among Swedish men

PONE-D-20-20129R1

Dear Dr. Laskowski,

We’re pleased to inform you that your manuscript has been judged scientifically suitable for publication and will be formally accepted for publication once it meets all outstanding technical requirements.

Kind regards,

Sreeram V. Ramagopalan

Academic Editor

PLOS ONE
---

## [Editor Report · Acceptance letter]

25 Nov 2020

PONE-D-20-20129R1 

Cardiorespiratory fitness in late adolescence and long-term risk of psoriasis and psoriatic arthritis among Swedish men 

Dear Dr. Laskowski:

I'm pleased to inform you that your manuscript has been deemed suitable for publication in PLOS ONE. Congratulations! Your manuscript is now with our production department. 

Kind regards, 

on behalf of

Dr. Sreeram V. Ramagopalan 

Academic Editor

PLOS ONE